# Longitudinal evidence that Event Related Potential measures of self-regulation do not predict everyday goal pursuit

Blair Saunders [1,5 ✉], Marina Milyavskaya [2,5 ✉] & Michael Inzlicht [3,4]

Self-regulation has been studied across levels of analysis; however, little attention has been paid to the extent to which self-report, neural, and behavioral indices predict goal pursuit in real-life. We use a mixed-method approach ($N = 201$) to triangulate evidence among established measures of different aspects of self-regulation to predict both the process of goal pursuit using experience sampling, as well as longer-term goal progress at 1, 3, and 6-month follow-ups. While self-reported trait self-control predicts goal attainment months later, we observe a null relationship between longitudinal goal attainment and ERPs associated with performance-monitoring and reactivity to positive/rewarding stimuli. Despite evidence that these ERPs are reliable and trait-like, and despite theorizing that suggests otherwise, our findings suggest that these ERPs are not meaningfully associated with everyday goal attainment. These findings challenge the ecological validity of brain measures thought to assess aspects of self-regulation.

[1] Psychology, School of Social Sciences, University of Dundee, Dundee, UK. [2] Department of Psychology, Carleton University, Ottawa, Canada. [3] Department of Psychology, University of Toronto, Toronto, Canada. [4] Department of Psychology and Rotman School of Management, University of Toronto, Toronto, Canada. [5] These authors contributed equally: Blair Saunders, Marina Milyavskaya. ✉email: b.z.saunders@dundee.ac.uk; marina.milyavskaya@carleton.ca

Goal progress is seldom straightforward. Mistakes, competing priorities, distractions, and temptations are common setbacks. The volatility of goal progress necessitates internal regulatory systems to represent our intentions, monitor for conflicting events, and exert control to create behavior that is aligned with our intentions. These systems are commonly identified as self-regulation, and typically involve a discrepancy-reducing process that reduces the distance between intended and actual states[1,2].

Self-regulation is the focus of intense investigation in psychology and neuroscience, and is variously identified as self-regulation, self-control, executive functioning, and/or cognitive control. This terminology has largely arisen in different sub-disciplines, and often co-occurs with differences in both methods and level of analysis, ranging from self-reported traits and reported everyday experience, to behavioral and neuroscientific analyses of speeded tasks. Despite obvious differences, each perspective agrees that self-regulation is not unitary, but that it instead relies on multiple processes, including goal setting, planning, reward sensitivity, performance monitoring, and inhibitory control, among others[2,3]. Each perspective also addresses a common overarching problem: how do we guide our thoughts, feelings, and behaviors out of trouble and towards an intended outcome? Testing this core question requires both reliable measurement of the various components of self-regulation, as well as high-quality tests of their predictive validity for everyday goal pursuit.

Some approaches to trait self-regulation have long grappled with these questions. Statistically powerful investigations have established the predictive validity of scale measures of self-regulation (e.g., trait conscientiousness, grit, and trait self-control) across multiple domains, including health behavior, academic attainment, morbidity, and mortality[4–10]. While theoretical differences exist among these traits, they also show considerable conceptual overlap and correlate strongly with each other ($rs > 0.7$; refs. [8,11]). In short, these traits lie within a conceptual space that behaves as self-regulation should; they correlate with other measures of self-regulation and predict real-world outcomes.

Experimental psychologists and cognitive neuroscientists have also conducted numerous laboratory experiments that propose to reveal the mental and biological processes underlying the various components of self-regulation. This research has been incredibly generative, revealing a range of highly replicable behavioral phenomena, computational models, and neural correlates of self-regulation that are associated with sensitivity to reward, detecting the need for self-regulation (e.g., performance-monitoring in the anterior midcingulate cortex, aMCC), and signaling to other brain areas to increase goal-directed top-down control[12–14].

Many theorists—including ourselves—have enthusiastically incorporated cognitive neuroscience methods to shed light on the mechanisms underlying self-regulation[15–18]. These approaches suggest that the neural mechanisms underlying the various components of cognitive control observed in the laboratory might predict self-regulation outside the lab. While numerous studies have investigated the brain as a predictor of real-world self-regulation [e.g., refs. [14,15,19–21]], not one study has examined the predictive validity of these measures in the real world longitudinally beyond a week or two. We tested this premise in the current study. Specifically, we tested the predictive validity of trait and neural measures associated with various aspects of self-regulation in a large sample that combined measures of multiple components of self-regulation across diverse methods, including EEG, behavioral tasks, ecological momentary assessment of self-regulatory processes (e.g., desire frequency, intensity, and resistance), and longer-term goal progress assessed at 1-, 3-, and 6-month follow-ups.

We investigated three event-related potentials (ERPs) that have been mapped onto components of self-regulation in which one top-down control system is capable of (down)regulating automatic processes that arise from a bottom-up habitual or reward-driven system [e.g., refs. [12,22]]. First, the error-related negativity (ERN[23]) is a response-locked ERP that differentiates error from correct responses within 100 ms and has been localized to the anterior midcingulate cortex[24]. The ERN might be one of the most replicable effects in all of cognitive neuroscience, and its relationship to internal performance-monitoring has led several theorists to directly implicate the ERN as a marker of a neurocognitive process underlying self-regulation in its broadest sense[15,16,18]. Empirical investigations of the real-world impact of the ERN suggest it might predict broad self-regulatory outcomes in the real world [e.g., refs. [21,25]].

ERP research has also revealed components related to the processing of appetitive stimuli. The reward positivity (RewP) arises 250–350 ms after feedback stimuli, and is potentiated to reinforcing signals[26]. The Late Positive Potential (LPP[27] is a positive ERP that develops over several seconds at parietal midline electrode sites from 300 ms after stimulus onset is maximal when highly arousing, motivationally relevant images are presented[28].

Unlike the ERN, which signals when control is needed, the RewP and LPP to positive images are better aligned with the appetitive processes that could undermine self-regulation (i.e., temptation, desire) if they come into conflict with a currently pursued goal[22]. That is, the RewP and LPP to positive images might reflect how reward sensitive a person is, and thus how motivated they are by appetitive stimuli. The ERN, in contrast, might reflect how attuned a person is to potential conflicts with their longstanding goals, including conflicts brought about by their responses to temptations in their environments.

Bridging the gap between laboratory and everyday self-regulation, and doing so longitudinally, allows us to test that the neural correlates of the various components of self-regulation elicited by laboratory tasks do not fall foul of several validity challenges. First, while well-controlled laboratory paradigms might allow for causal tests of predictions made from theories, internal validity sometimes comes at a cost to external validity[29,30]. The so-called mutual internal validity problem can become particularly acute if outcomes from lab experiments are used to develop theories whose predictions are tested through an iterative process of further lab experimentation that eventually focuses on the explanation of artificial lab tasks at the expense of ecological validity[31]. As a result, established lab tasks can become unmoored from the reality they are trying to model.

Neural measures that are commonly related to aspects of self-regulation might suffer from the mutual internal validity problem. After all, they are often studied in tightly controlled laboratory behavioral tasks (e.g., the Stroop, Flanker, or Go/no-go task) that were designed for the detailed examination and explanation of the behavioral, neural, and computational correlates of cognitive control [e.g., refs. [12,32,33]] rather than as predictors of real-world outcomes. Avoiding the mutual internal validity problem in the current context requires testing the ability of various putative neural correlates to predict self-regulation outcomes outside the laboratory.

Partial support for the candidacy of the ERN, RewP, and LPP as predictors of real-world outcomes comes from demonstrations that ERP scores typically possess psychometric properties (i.e., internal consistency, heritability, test-retest reliability) that are consistent with stable individual differences[34–39]. While such reliability is only a precondition for establishing construct validity, it is noteworthy that metrics derived from laboratory behavioral measures often show relatively poor reliability[40,41].

This difference in reliability between scores on neural and behavioral measures of aspects of self-regulation, then, opens the possibility that while task measures have limited real-world predictive value[42,43], the ERPs derived from the very same tasks are nevertheless plausible longitudinal predictors.

The validity of these ERPs as neural correlates of aspects of self-regulation also depends on the extent to which they behave as trait self-regulation should. This can be established by testing their relationship to a network of other measures that sit within the conceptual space occupied by self-regulation[44]. In short, these ERP components should relate to other trait measures related to self-regulation (i.e., convergent validity) as well as to outcomes that theoretically result from good self-regulation.

There is mixed support for the suggestion that the ERN sits well within this broad view of self-regulation. Many tests have demonstrated that the ERN is related to outcomes such as academic attainment, obesity, and everyday emotion regulation[25,45–47], all of which might reflect good self-regulation (among other things). The component also shows lower amplitude in groups associated with a range of self-regulation problems, including addictions[48–51]. Even though the ERN relates to one aspect of self-regulation (error monitoring), this aspect is thought to be at the very core of the self-regulation process[15]. It is why the numerous previous studies have sought to associate it with broad self-regulatory outcomes with no direct mechanistic relationship with error monitoring.

However, there is little evidence that the ERN is correlated with self-regulatory personality traits. Instead, increased ERN amplitudes are commonly associated with anxious psychopathology and neuroticism[52–55]. These findings question the positioning of the ERN within the traditional conceptual space of self-regulation, as good self-regulation is commonly associated with increased psychological adjustment[9] and satisfaction with life[56]. These factors prompt some caution regarding a sweeping hypothesis that larger ERNs necessarily predict better self-regulation, although it is possible that the ERN is linked to self-regulation through separate mechanisms that involve the integration of negative affect and neural monitoring[57–59].

Both the RewP and LPP seem to fit more conventionally within a network of broad self-regulation given their clear association with reward sensitivity and approach motivation. The LPP is elevated to positive images for multiple groups with substance use disorders[60–62], extroversion[63,64], and trait behavioral approach[65]. The RewP is positively correlated with both subjective liking of rewards[66], reward sensitivity[67], trait approach motivation[68], and extroversion[69]. Both reward sensitivity and approach motivation are strongly implicated in trait impulsivity[70], suggesting that people who have high RewPs or LPPs might struggle to control their impulses, and struggle with self-regulation.

One last factor that urges caution when making predictions from the existing literature is low statistical power that is known to inflate published effects as well as increase rates of both false positives and false negatives[71]. Underpowered studies are a challenge for many disciplines; however, recent estimates suggest that cognitive neuroscience might be especially underpowered to detect all but unrealistically large effect sizes[72–75]. Large effect sizes are generally implausible across psychology[76], and are particularly unlikely in the current case when a relatively narrowly defined measure (e.g., neural reactivity to mistakes in a flanker task) would be unlikely to explain large amounts of variance in noisy real-world outcomes [cf., ref. [77]]. Thus, studies investigating real-world prediction, we believe, should be powered to detect small-to-medium effect sizes.

While self-report measures are highly scalable, neuroscience is considerably more resource intensive[78]. Indeed, ERP sample sizes tend to be smaller for frequently studied individual differences (e.g., mean N = 66 in a meta-analysis of anxiety-ERN relationship,

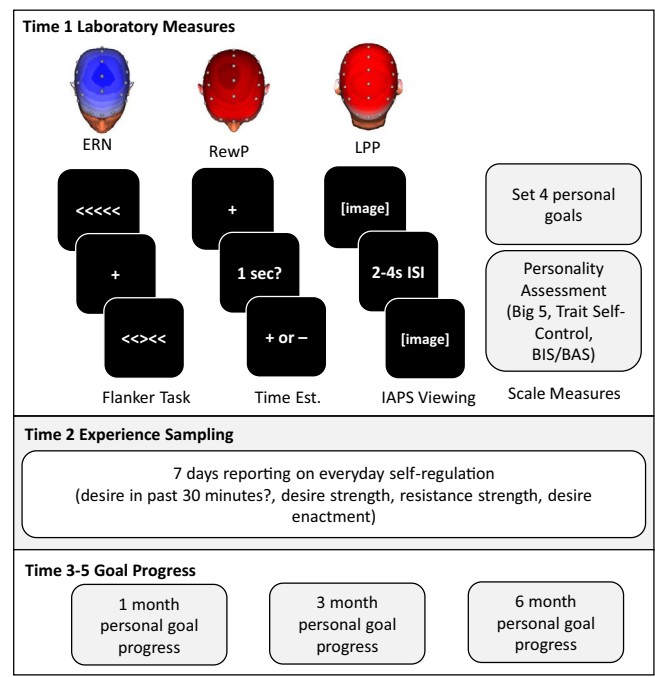

**Fig. 1 Schematic overview of our longitudinal mixed-methods design.** The laboratory measures were assessed at intake (top panel), the subsequent week-long experience sampling at time two (middle panel), and the three follow-ups on personal goal progress at time-points 3–5, corresponding to 1-, 3-, and 6-month follow-ups, respectively. ERN: error-related negativity; RewP: reward positivity; LPP: late positive potential; ISI: inter-stimulus interval; BIS: Behavioral Inhibition System; BAS: Behavioral Activation System; IAPS: International Affective Picture System.

ref. [54]), and studies that have tested the predictive validity of the ERN have returned mixed results [e.g., refs. [21,25,79]]. Thus, the existing evidence provides inconsistent support for the real-world predictive validity of the ERN, often in studies that are likely underpowered to detect small-to-medium effect sizes, do not include longer-term follow-ups (i.e., beyond a few weeks), and/or were not preregistered. We aimed to provide a well-powered test of the predictive validity of the ERN in the context of a wide conceptualization of self-regulation including scale trait measures, everyday experience sampling, and the longitudinal attainment of personal goals.

Here, we took a mixed-method approach to explore the predictive validity of self-report (trait self-control, conscientiousness, behavioral approach system), neural (ERN, RewP, LPP), and behavioral task (cognitive control on the flanker task) measures of aspects of self-regulation as predictors of real-world goal processes unfolding both in the moment (assessed through experience sampling) and over longer (1-, 3-, and 6-month) periods (see Fig. 1). Critically, we preregistered a number of our hypotheses in advance of data analysis.

Our mixed-method design and large sample size also allowed us to describe the relationship broadly among a range of neural, behavioral, and self-report assessments of aspects of self-regulation. This had multiple elements, including exploring convergence among diverse self-regulation measures, testing for the predictive validity of each measure as a determinant of long-term goal attainment, and aiming to replicate previous relationships (e.g., that temptation and not self-control predicts goal success, ref. [80]). However, the central focus of this paper relates to the relationship between ERPs and real-world self-regulation (i.e., experience sampling, long-term goal attainment). The ERP hypotheses were preregistered (https://osf.io/v8jzd/) as follows:

**Table 1 Descriptive statistics and correlations for baseline personality and EEG measures.**

| | 1 | 2 | 3 | 4 | 5 | 6 | 7 | 8 | 9 | 10 | 11 | 12 | 13 | 14 | 15 | 16 |
|---|---|---|---|---|---|---|---|---|---|---|---|---|---|---|---|---|
| Valid N | 201 | 201 | 201 | 201 | 201 | 201 | 201 | 201 | 192 | 190 | 190 | 170 | 189 | 189 | 181 | 181 |
| Mean | 3.21 | 3.76 | 3.36 | 3.02 | 3.58 | 4.05 | 2.95 | 3.12 | 6.39 | 2.93 | −0.08 | −10.44 | 5.36 | 11.03 | 87.44 | 0.17 |
| Std. deviation | 0.86 | 0.64 | 0.64 | 0.81 | 0.52 | 1.02 | 0.56 | 0.41 | 5.26 | 2.88 | 2.72 | 6.66 | 4.41 | 7.96 | 24.87 | 0.11 |
| Range | 1.13–5.00 | 1.78–4.89 | 1.78–5.00 | 1.13–4.88 | 2.20–5.00 | 1.77–6.00 | 1.00–4.00 | 2.15–4.00 | −7.25–22.4 | −8.05–11.75 | −19.88–7.21 | −34.90–2.32 | −3.70–17.50 | −4.52–30.16 | 18.34–194.63 | −0.03–0.52 |
| S.E.M. | 0.06 | 0.05 | 0.05 | 0.06 | 0.04 | 0.07 | 0.04 | 0.03 | 0.38 | 0.21 | 0.20 | 0.51 | 0.32 | 0.58 | 1.85 | 0.01 |
| Reliability | 0.87 | 0.76 | 0.76 | 0.83 | 0.65 | 0.84 | 0.78 | 0.81 | 0.82 | 0.54 | 0.23 | 0.89 | 0.80 | 0.97 | — | — |
| 1. Extraversion | — | | | | | | | | | | | | | | | |
| 2. Agreeableness | 0.12 | — | | | | | | | | | | | | | | |
| 3. Conscientiousness | 0.14* | 0.30*** | — | | | | | | | | | | | | | |
| 4. Neuroticism | −0.19** | −0.22** | −0.18** | — | | | | | | | | | | | | |
| 5. Openness | 0.29*** | 0.14* | 0.16* | 0.07 | — | | | | | | | | | | | |
| 6. TSC | 0.07 | 0.28*** | 0.69*** | −0.30*** | 0.10 | — | | | | | | | | | | |
| 7. BIS | −0.16* | 0.13 | −0.05 | 0.66*** | 0.07 | −0.17* | — | | | | | | | | | |
| 8. BAS | 0.49*** | 0.09 | −0.01 | −0.03 | 0.39*** | −0.12 | 0.07 | — | | | | | | | | |
| 9. Early LPP high arousal pleasant | 0.04 | 0.09 | −0.07 | 0.02 | 0.05 | −0.02 | 0.05 | 0.03 | — | | | | | | | |
| 10. LPP difference high – low arousal | 0.06 | 0.02 | −0.04 | 0.01 | 0.06 | −0.02 | 0.13 | 0.06 | 0.43*** | — | | | | | | |
| 11. LPP difference: positive – negative | −0.08 | −0.08 | −0.06 | 0.07 | 0.03 | 0.00 | 0.00 | −0.04 | 0.13 | −0.00 | — | | | | | |
| 12. ΔERN | 0.09 | 0.14 | −0.06 | 0.09 | 0.01 | −0.02 | 0.14 | 0.07 | 0.11 | 0.09 | −0.08 | — | | | | |
| 13. ΔRewP | 0.03 | 0.04 | −0.10 | 0.02 | 0.04 | −0.03 | 0.07 | 0.08 | 0.20** | 0.18* | 0.01 | 0.02 | — | | | |
| 14. RewP | −0.09 | 0.04 | −0.04 | −0.02 | 0.08 | −0.04 | 0.06 | 0.12 | 0.16* | 0.15* | 0.06 | −0.06 | 0.57*** | — | | |
| 15. Flanker ΔRT | 0.09 | −0.10 | −0.02 | 0.02 | −0.08 | −0.06 | 0.03 | −0.16* | 0.01 | 0.07 | −0.14 | 0.03 | 0.03 | −0.03 | — | |
| 16. Flanker Δ error rate | 0.09 | −0.10 | −0.07 | 0.01 | −0.05 | −0.06 | −0.07 | −0.03 | 0.02 | 0.13 | −0.05 | 0.36*** | 0.01 | −0.05 | 0.20** | — |

All correlation coefficients (i.e., rows numbered 1–16) depict Pearson's correlations (two-tailed). *p < 0.05, **p < 0.01, ***p < 0.001, exact p-values above 0.001 and 95% CIs for correlations are available online: https://osf.io/39rt5/. Reliability = Cronbach's alpha (measures 1–8) and split-half reliability (measures 9–14).
TSC trait self-control, BIS behavior inhibition, BAS behavior approach, LPP late positive potential, ΔERN difference ERN, ΔRewP difference reward positivity, RT reaction time, S.E.M. standard error of the mean.

1. ERPs related to the processing of positive/appetitive stimuli (i.e., the LPP and RewP) will be associated with reduced long-term goal progress.

2. The LPP and RewP will be associated with in-the-moment experiences indicative of poor self-regulation (increased desires, increased enactment, reduced resistance). If significant, momentary self-regulation variables will be tested as mediators of the relationship between the LPP/RewP and goal progress.

3. Based on prior findings that the ERN relates to everyday self-regulation, we predicted that higher ERN amplitudes should relate to greater goal progress.

4. A self-regulation account of the ERN also suggests that the component will be associated with enhanced self-regulation (i.e., reduced desires, reduced enactment, enhanced resistance). If significant, these relationships will be tested as mediators of the relationship between the ERN and goal progress.

5. Hypotheses regarding the ERN were presented with the caveat that this component is often associated with forms of anxiety/negative affectivity that are often inversely related to self-regulation. We preregistered our ambivalence about the ERN-regulation relationship, suggesting that this component may be unrelated to momentary self-regulation and long-term goal attainment.

## Results

**Descriptive statistics and convergence among self-regulation measures.** Table 1 presents all the descriptive statistics and intercorrelations for the personality, EEG, and cognitive measures assessed at intake. Figures 2–4 show the canonical ERP effects for the ERN, RewP, and LPP. Further analyses of our ERP results and cognitive task performance are presented in the Supplementary Information (also available on the Open Science Framework: https://osf.io/kmj9w/). As can also be seen in Fig. 5, self-report measures related to self-regulation (conscientiousness and trait self-control) were strongly correlated with each other. The neural measures, however, were uncorrelated with self-reported personality traits. The RewP and LPP were also uncorrelated with a self-report measure of behavior activation (the Behavior Activation Scale; BAS).

Many potential variables could be extracted from ERP averages to associate with aspects of self-regulation. We selected 6 ERPs to represent theoretically relevant constructs while avoiding multiple comparisons including redundant, highly correlated variables (e.g., including the ERN, CRN, and ΔERN). The difference ERN (ΔERN) was chosen as a measure reflecting the sensitivity of a monitoring system that differentiates between error and correct trials. Two variables represented neural responses to feedback; the RewP to capture neural reactivity to positive reinforcement, and the ΔRewP to reflect the feedback monitoring systems relative reactivity to feedback valence (correct-error). For the LPP, we included the early LPP to positive stimuli to capture initial orienting to appetitive stimuli, as well as two difference waves on the entire LPP window (early and late) that reflected the arousal (high-low) and valence (positive-negative) dimensions of affect[81]. We also conducted various exploratory analyses using alternate operationalizations of the ERPs, never finding results in disagreement with our main conclusions (see https://osf.io/kmj9w/).

Table 2 presents the descriptive statistics and intercorrelations for the experience sampling and goal progress variables. On average, 60% of participants' desires conflicted with at least one goal. As in previous research[80,82], greater resistance was related to reduced enactment of a desire, at least in the moment. However,

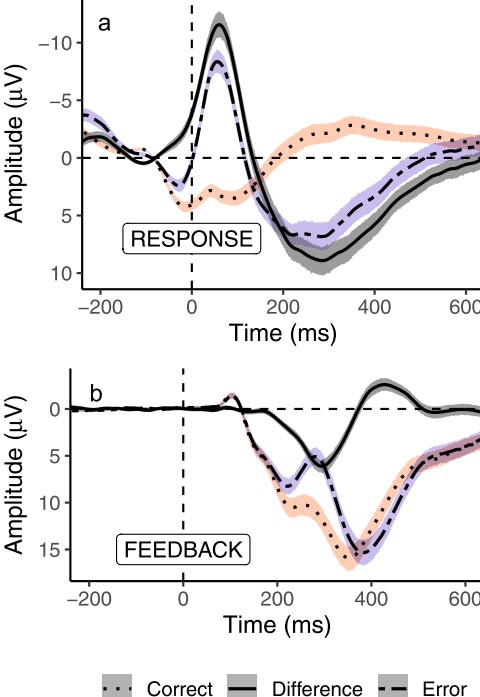

**Fig. 2 The ERN and RewP at electrode FCz. a**, **b** Central lines depict the grand average ERP amplitude across participants (dotted line: correct; dash-dotted line: error; solid line: difference wave, and the shaded error bands denote 95% between-subjects confidence intervals (red: correct; blue: error; black: difference wave). Timepoint zero in the ERN waveform refers to the participants button-press response, whereas timepoint zero in the RewP waveform refers to the onset of the external feedback stimulus. The ERN was significantly more negative on error than correct trials, two-sided paired samples Student's $t$ test: $t(169) = 20.42$, $p < 0.001$, $d = 1.57$, 95% CIs [1.34, 1.79], and the RewP was significantly more positive to correct feedback than error feedback on correct trials, two-sided paired samples Student's $t$-test: $t(188) = 16.73$, $p < 0.001$, $d = 1.22$, 95% CIs [1.03, 1.40].

unlike in past research, greater desires were not significantly related to goal progress months later. Surprisingly, enactment had a small, positive relationship with goal progress at 1 and 3 months, suggesting that those who gave in to their desires reported more goal progress at later intervals. As in previous research[80], experiencing greater conflict with personally important goals (i.e., more temptations) was related to lower progress on those goals months later; stronger resistance, on the other hand, was unrelated to goal progress. Additional analyses were conducted examining conflicting desires only (looking both at desires conflicting with goals, and desires which are at least somewhat resisted (i.e., problematic)); results from these analyses do not change any of our conclusions (see Table S2 in the Supplementary Information). Finally, we conducted additional analyses examining our preregistered hypothesis regarding desire and resistance as predictors of goal progress and report them in the Supplementary Information (see Table S1).

**Relations with ESM**. Table 3 presents the correlations along with corrected and uncorrected $p$-values, and Bayes factors; Fig. 5 shows the heat map associated with that table. Consistent with previous research, trait self-control was related to lower average resistance; the higher people's trait self-control, the less they relied on self-control in the moment. However, that correlation was not robust to multiple comparisons or the Bayes factor

analyses. Only two correlations remained significant when controlling for multiple comparisons: both agreeableness and BAS were related to experiencing stronger experienced desires. Exploratory analyses with conflict strength showed that neuroticism was related to greater conflict and openness to experience with lower conflict. None of our a-priori hypotheses regarding neural correlates of desire were supported. Furthermore, looking at Bayes factors for those correlations, the null hypotheses were 4–10 times better than the alternatives, suggesting moderate evidence in favor of the null hypotheses.

**Relation with goal progress**. Finally, we examined preregistered hypotheses related to goal progress. Table 4 presents the correlations along with corrected and uncorrected $p$-values, and Bayes factors; Fig. 5 shows the heat map associated with that table. As expected, trait self-control was related to goal progress at all time points (although the relation was non-significant at the second follow-up after we corrected for multiple comparisons; see also Fig. 6). Surprisingly, conscientiousness was not significantly related to progress (despite a strong correlation with trait self-control). Unexpectedly, agreeableness was positively related to goal progress and neuroticism negatively related to goal progress at two follow-ups. None of the neural indicators nor behavioral indicators were related to goal progress. This is contrary to our hypotheses predicting a relationship between RewP and goal progress and between the LPP and goal progress, but in line with our competing prediction regarding ERN and behavioral task measures, where we pre-registered competing predictions for both these variables. Bayes Factors show that for neural indicators, the evidence in favor of the null is moderate (3 to 9 times stronger than for the alternate). For behavioral measures extracted from the flanker task, the evidence in favor of the null is weak to moderate ($BF_{01}$ of 1.3 to 9.1). Note that we preregistered an examination of whether in-the-moment desire and resistance mediate potential relations between personality/neural indicators (at baseline) and goal progress. However, given that there are no main effects or relations between self-regulatory variables and goal progress, we did not conduct those analyses.

## Discussion

We investigated diverse neural, behavioral task, and self-report measures broadly related to various aspects of self-regulation as predictors for everyday goal pursuit both during in-the-moment goal pursuit (i.e., during a week of experience sampling) and longitudinally in the way of 1-, 3-, and 6-month assessments of personal goal progress. None of the neural indicators (RewP, LPP, nor ERN) were related to self-reported traits, experienced desires, desire resistance, or long-term goal progress. In fact, Bayes Factors indicated that it was 4–10 times more likely that neural indicators were not meaningfully related to either momentary or long-term self-regulation. These results are consistent with widespread jingle-fallacies in self-regulation research where various measures labeled as related to aspects of self-regulation are largely uncorrelated with each other. In addition, we found variation in predictive validity among self-regulation measures. While our ERP measures were unrelated to other assessments of self-regulation, trait self-control predicted greater goal progress up to six months later, and trait behavioral approach was associated with subsequent desire intensity during experience sampling.

Here, we present a large preregistered study assessing the predictive validity of ERPs related to performance monitoring (ERN), feedback processing (RewP), and motivated attention to external stimuli (LPP) in the context of longitudinal goal attainment. Neural reactivity to positively-valenced events was not associated with a higher prevalence or intensity of daily desires during experience

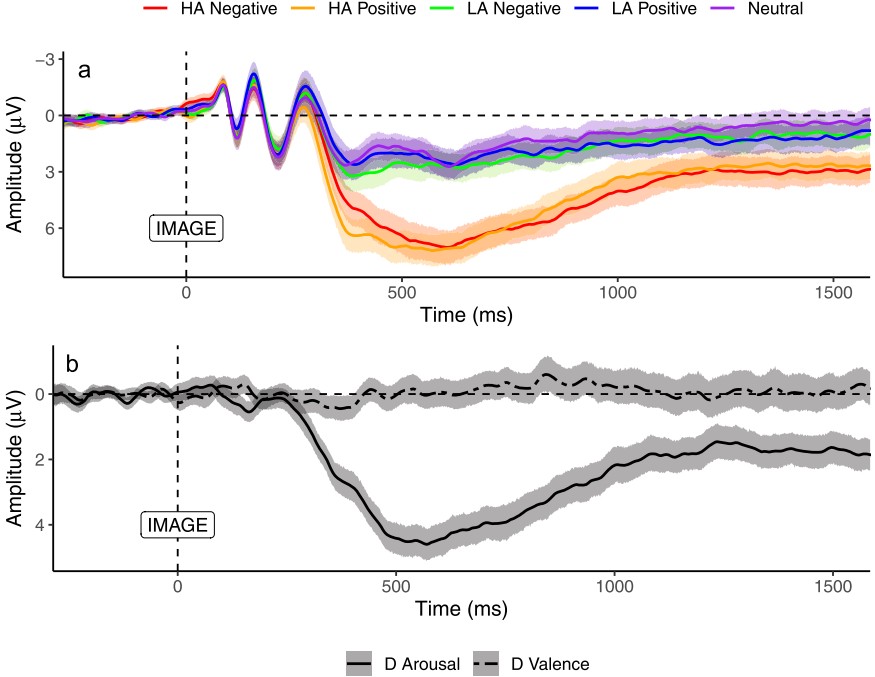

**Fig. 3 Figure depicting LPP at electrode Pz.** Central lines depict the grand average ERP amplitude across participants, and the shaded error bands denote 95% between-subjects confidence intervals (red: high arousal negative; orange: high arousal positive; green: low arousal negative; blue: low arousal positive; purple: neutral). **a** ERPs showing the LPP for every condition in the experiment. A 2 (valence: positive vs. negative) × 2 (arousal: high vs. low) × 2 (window: early LPP vs. late LPP) repeated-measures ANOVA was used to assess the LPP across image types revealed a significant three-way interaction, F(1, 189) = 23.34, p < 0.001. **b** ERPs showing the effect of valence (positive minus negative—dashed black line) and arousal (high minus low—solid black line) on the LPP amplitude. HA, high arousal image; LA, low arousal image; D denotes difference wave.

sampling, or to long-term goal attainment. The ERN was also not associated with reduced enactment of desires or with eventual goal attainment. There are strong theoretical reasons to believe the ERN should predict real-world goal attainment—after all, it is only via continuous monitoring of one's behavior that discrepancies between longstanding goals and current desires can be noted and resolved[15]. Our null results provide a cautionary note to previous theorists, including ourselves, who have put forward the ERN as an analog to the self-monitoring processes emphasized by traditional cybernetic models of self-regulation [e.g., refs. [15,18]]. Our results do not challenge the role of the ERN within existing cognitive neuroscience models (e.g., the conflict monitoring account[12]), but do suggest that the ERN is potentially often mischaracterized as a measure of self-regulation. Given the common conceptualization of self-regulation as a psychological construct that predicts human health, wellbeing, and goal attainment[2], our results suggest that the ERN does not behave as a measure of self-regulation should.

Our ERP results complement suggestions that behavioral measures of cognitive control do not predict the types of self-regulation that are relevant for everyday life[11,43]. While basic psychometric limitations might explain the null associations between behavioral tasks and everyday self-regulation[40–42], our ERPs demonstrated good-to-excellent internal consistency. Thus, the non-convergence points more directly to problems in the conceptualization of these variables themselves as measuring aspects of self-regulation. Scale measures ask individuals to report how they generally act across a spectrum of regulation-relevant items (e.g., "I am good at resisting temptation"; "I am lazy"). Thus, the broader bandwidth of these scales could account for the relation we observe between trait self-control and goal attainment (see ref. [77]). Conversely, ERPs were assessed in a one-off laboratory setting largely dissimilar to context in which everyday control occurs. Perhaps ERPs, then, measure a

narrower, less ecologically valid construct than self-report measures. Such considerations have not prevented scholars from exploring whether ERPs self-regulation-related outcomes; but we now wonder if such examinations were asking these ERPs to predict far more than they are capable of.

Previous work has found mixed support for the relationship between the ERN and various outcomes [e.g., refs. [21,25,79]]. Direct comparison with previous studies is difficult because of several methodological differences. That said, we did operationalize the ERN in a traditional manner and our study features a large sample, preregistered analysis plan, and multiple longitudinal follow-ups. Thus, if the ERN was predictive of personal goal pursuit, our design should have found evidence supporting this hypothesis. Our ERN results were consistent across measures; however, this consistency was in the form of repeated support for a null relationship between the ERN and all other measures that could be identified as either directly related to established measures of self-regulation (e.g., trait self-control, daily resistance) or as an expected outcome of successful self-regulation (reduced enactment, attainment of personal goals). While no single study can provide a definitive answer to the broadest question of the general predictive validity of the ERN, our results (in addition to considerations of the likely low statistical power of past investigations; ref. [83]) raise questions about the ability of lab-derived ERN scores to predict the longer-term pursuit of personal goals. Future work would benefit from conducting larger-scale confirmatory attempts to replicate previously identified ERN-outcome associations in other, perhaps more specific, domains (e.g., between the ERN and academic grades, or between the ERN and everyday emotion regulation).

It is worth noting that individual differences in ERN amplitudes often fail to predict between subject variation during the

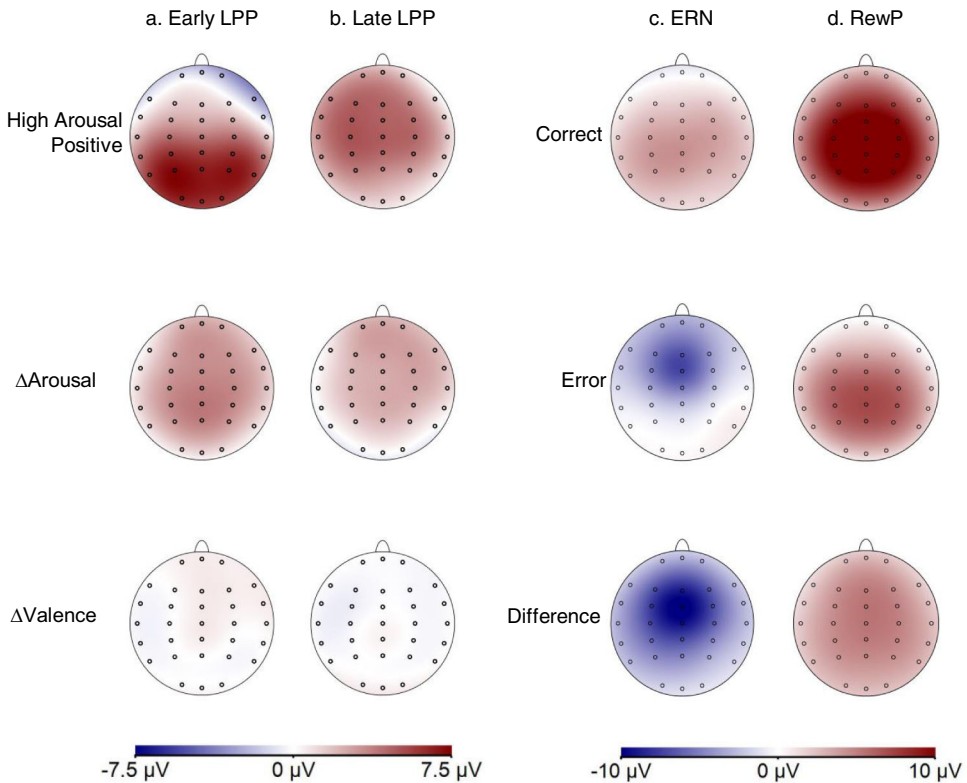

**Fig. 4 Topographic maps showing the scalp distribution of the grand average ERPs.** Increasing red intensity depicts positive-going ERP amplitudes, blue intensity depicts negative ERP amplitudes. **a** The early LPP for high-arousal positive images (top), the difference score for high arousal low arousal (middle), and the difference score for positive valence minus negative valence (bottom). **b** The equivalent topographic maps as in (**a**), but for the late LPP window. **c** Topographic maps in the time course of the ERN for correct trials (top), error trials (middle), and the difference score subtracting correct trials from errors (bottom). **d** The same pattern of topographic maps as in (**c**), but this time for the time course of the RewP after feedback stimuli. LPP: late positive potential; ERN: Error-related negativity; RewP: reward positivity.

performance of laboratory cognitive control tasks themselves, such as slowing after initial mistakes to become more cautious (i.e., posterror slowing[84]). These findings, in addition to our own, suggest that the ERN might serve poorly as an individual difference predictor of subsequent behavior—even when this behavioral adjustment and the ERN arise within the context of the same task. In contrast, within-person variation in ERN amplitude does predict within-person adjustments in response caution on a trial-by-trial basis[84,85]. Future studies might test this idea outside of the lab by, for example, pairing ambulatory assessments of the ERN with experience sampling to test if within-person error-related brain activity tracks with enhanced self-regulation on a moment-to-moment basis.

While trait self-control was associated with long-term goal progress, we found surprisingly few correlations between most self-reported traits and both momentary experience sampling and longer-term goal attainment. One exception was the BAS, which was positively related to greater desire strength, replicating previous work[82,86]. Unlike past research [e.g., refs. [82,87]], trait self-control was not significantly related to frequency of desire or desire strength; for resistance and conflict there was a relationship in the expected direction (more self-control related to less resistance and less conflict), but it was not robust to correction, and Bayes Factor (of 1.0 or less) showed no support either for or against it. We also found virtually no relations between big-five personality and momentary measures related to aspects of self-regulation. One other study reported on the interaction between prior self-control and each of the big five on desire enactment, but

did not report main effects[88]. Prior research on state manifestation of personality finds that in typical behavior there is a lot of within-person variability, such that resisting desires may not be a behavioral manifestation of conscientiousness[89]. We were also surprised that agreeableness was related to experiencing stronger desires; future research needs to replicate this finding and explore it further.

Past research suggests that most of the variance in goal progress is at the goal level (i.e., within-person; see ref. [90], for an overview), perhaps explaining the few associations between individual differences and goal attainment. Besides self-control, agreeableness was also positively related to goal progress, and neuroticism negatively related to progress (at 2 out of the 3 follow-ups). These were exploratory results (not preregistered), but they do replicate some past research on these traits and personal goal progress[91–93]. Conscientiousness was unrelated to goal progress, adding to the mixed literature on the role of conscientiousness in personal goal pursuit[91,92,94,95].

Our work suggests several avenues for future research. We examined self-regulation in the real world by examining goal pursuit, broadly construed. ERPs might predict outcomes in the real world, but perhaps only in those specific outcomes that closely resemble the processes thought to be tapped by these ERPs. Future studies, therefore, might benefit from examining error-monitoring or reward-monitoring outside the lab (e.g., probing awareness of errors or reward) and then investigating whether this awareness is correlated with their attendant ERPs. Building on recent suggestions that self-regulation resembles

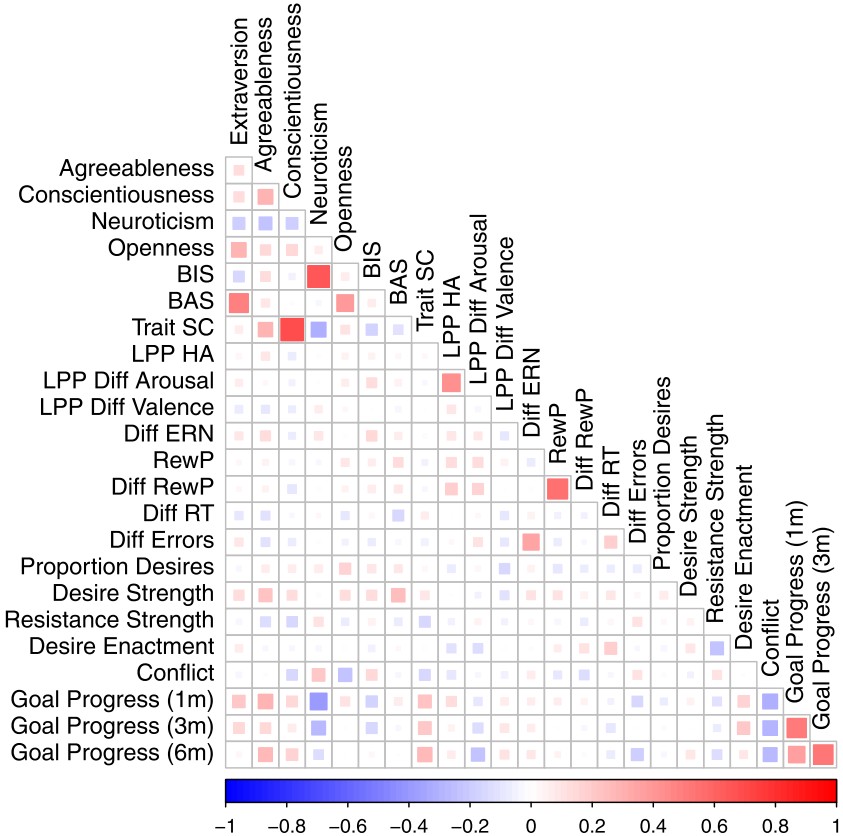

**Fig. 5 Pearson's correlation plots showing associations among dependent variables in our study.** Increasing size and saturation of squares depict the effect size, negative correlations are shown in blue and positive correlations in red. BIS: Behavioral Inhibition System; BAS: Behavioral Activation System; Trait SC: Trait Self-Control Scale; LPP HA: LPP to high arousal positive images; LPP Diff Arousal: LPP difference score (high arousal − low arousal); LPP Diff Valence: LPP difference score (positive valence − negative valence); Diff ERN: ERN difference score (error − correct); RewP: Reward Positivity; Diff Rew P: RewP difference score (correct − error); Diff RT: reaction time difference on the flanker task (incompatible–compatible); Diff Errors: error rate difference on the flanker task (incompatible–compatible).

value-based choice[96], one fMRI study recently found that the neural correlates of subjective valuation processes did predict everyday self-regulation in a sample of almost 200 participants[97]. Thus, examining self-regulation in the real world by examining psychological variables more closely-aligned with theorized processes—in this case taking a decision-making approach to self-regulation [e.g., ref. [98]]—may be fruitful for future research.

In addition, even though experience sampling results in many observations per participant, here we aggregated these observations to examine individual differences (i.e., in how they generally feel and act across the week). This comes at a cost to power. Although our sample sizes are high for an EEG study, we only had 80% power to detect correlations greater than $r = 0.23$ at the one-month follow-up, rising to $r = 0.27$ at six months. As such, our study meets common power conventions to detect small effect sizes[99], but has lower power to detect even smaller effects. It is important to note that our Bayesian analyses did provide evidence in favor of null relationships—suggesting that samples were not too small to provide evidence in relation to our hypotheses. However, these considerations highlight that potentially even larger, preregistered studies are required to continue to investigate the predictive validity of ERPs for everyday outcomes. Increased statistical power will not, however, overcome the possibility (as discussed above) that individual differences extracted from behavioral cognitive control tasks, including their neural correlates, are not particularly valid predictors of everyday self-regulation.

In conclusion, for studies and theories to be meaningful, they must iteratively move between the laboratory and the real world to avoid generating elaborate theories and tasks that account for little variability in the real world, the so-called mutual internal validity problem[31]. Our results suggest that some research on self-regulation has fallen prey to this validity issue, casting doubt on the broad applicability of past research. Our findings provide evidence that a range of ERPs related to performance monitoring (i.e., the ERN) and neural reactivity to positive stimuli (RewP, LPP) are not associated with measures of self-regulation, including self-reported self-control, everyday desire, and resistance during experience sampling, or the long-term attainment of personally selected goals. These findings challenge the ecological validity of brain measures thought to assess aspects of self-regulation. These measures do not appear to predict goal-directed behavior in the real world and challenge a too simplified view of self-regulation.

## Methods

**Open science statement**. All hypotheses and analytical plans were registered on the Open Science Framework (https://osf.io/v8jzd/) after the data was collected but before anyone looked at the longer-term follow-up data (date of latest registration: 16 July 2020). Our original analysis plan also included exploratory analyses including time-frequency analyses in the theta and delta band. However, we opted to only conduct and report the confirmatory ERP analyses to reduce the complexity of our report. The exploratory analyses have not been conducted to date, and, as such, we cannot make any claims about the predictive/ecological validity of time-frequency approaches to studying cognitive control as a measure of self-regulation.

**Table 2 Descriptive statistics and correlations for experience sampling and goal progress variables.**

|  | 1 | 2 | 3 | 4 | 5 | 6 | 7 | 8 |
|---|---|---|---|---|---|---|---|---|
| Valid N | 193 | 192 | 192 | 191 | 180 | 149 | 132 | 107 |
| Mean | 64.02 | 5.24 | 2.81 | 0.47 | 0.2 | 3.73 | 4.06 | 4.25 |
| Std. deviation | 24.26 | 0.90 | 1.15 | 0.21 | 0.63 | 1.04 | 0.99 | 1.15 |
| Range | 0–100 | 1–7 | 1–7 | 0–1 | −2.30–1.88 | 1–7 | 1.25–7 | 1.83–7 |
| 1. Proportion of desires | — | | | | | | | |
| 2. Desire strength | 0.08 | — | | | | | | |
| 3. Resistance strength | 0.03 | 0.05 | — | | | | | |
| 4. Desire enactment | −0.04 | 0.10 | −0.23** | — | | | | |
| 5. Average conflict | −0.02 | −0.04 | 0.11 | −0.01 | — | | | |
| 6. Goal progress at 1 month | −0.06 | 0.06 | −0.11 | 0.17* | −0.30*** | — | | |
| 7. Goal progress at 3 months | −0.02 | 0.00 | −0.06 | 0.20* | −0.29** | 0.51*** | — | |
| 8. Goal progress at 6 months | −0.03 | 0.10 | −0.12 | 0.08 | −0.26** | 0.37*** | 0.53*** | — |

All correlation coefficients (i.e., rows numbered 1–16) depict Person's correlations (two-tailed), *$p < 0.05$, **$p < 0.01$, ***$p < 0.001$, exact $p$-values above 0.001 and 95% CIs are depicted online: https://osf.io/39rt5/.

**Participants and procedure**. Participants ($N = 226$) were predominantly recruited using a convenience sampling approach through an undergraduate participant pool, though a smaller number were also recruited through on-campus and local advertisements. Participants were compensated with up to $75CAD for participating in the study ($25 for the initial lab portion and the experience sampling, with an additional opportunity to earn up to a $20 for completing 85% + of the experience sampling signals, and $10 for completing 75–85%. Participants also received $5 for each follow-up, and a $5 bonus for completing all follow-ups). Participants could choose between receiving their compensation in cash or as an online shopping voucher. Participants were 37% male, 63% female with a mean age of =20.4 (SD = 5.93). 92.4% of the participants reported being a current student.

Although we did not conduct an a-priori power analysis, a sensitivity analysis showed that this sample size would allow us to find effects as small as $r = 0.19$ with 80% power for the between-subject analyses. Participants came into the lab for a 2-h session during which they completed questionnaires and computerized tasks while their brain activity was recorded with EEG. Three computerized tasks were administered, in addition to baseline measures of EEG activity: a flanker task (to assess error-related negativity, ERN), a passive image viewing task (to assess late positive potential, LPP), and a time estimation task (to assess reward positivity, RewP). All materials can be found at OSF. A week later, participants begun the experience sampling portion of the study: each day for seven days, participants received seven signals with brief surveys. Using SurveySignal[100], these signals were sent at random times in seven equal intervals between 9:30 am and 9:30 pm. Please note that we also collected additional data on in-the-moment self-regulation strategies, which were reported in Milyavskaya et al.[101]. As part of a larger data collection, we also collected data from a nightly survey sent at 10:15 pm each day of the experience sampling portion of the study. These questionnaires were included to answer questions beyond the scope of this paper and will not be reported further. Participants received online follow-ups 1 month, 3 months, and 6 months after their lab session. There was also another follow-up survey sent out 12 months after intake; however, we wanted to examine goal pursuit only for those goals that participants were still pursuing, and participants did not complete measures of goal progress for all their goals at the 12-month follow-up; as such, we did not include it in the present paper. Each follow-up survey piped in the goals that participants indicated in the intake survey and asked them a series of questions about goal pursuit, including a measure of goal progress. Participants provided informed consent, and we complied with a protocol that was reviewed and approved by the University of Toronto Research Ethics Board, Social Sciences, Humanities, and Education Committee (approval number: 30380).

Following data cleaning and participant non-response to follow-up surveys (see Supplementary Information: https://osf.io/g759u/), we were left with 201 participants at baseline (first survey and experience sampling), 149 at the 1-month follow-up, 132 at the 3-month follow-up, and 107 at the 6-month follow-up. Follow-up sensitivity analyses showed that these sample sizes were sufficient to detect effects as small as $r = 0.23$ at the first follow-up, $r = 0.24$ at the second follow-up, and $r = 0.27$ at the third follow-up with 80% power for the between-subject analyses.

**Materials**

*Personality*. Trait self-control was measured using the trait self-control scale[9], consisting of 13 items (e.g., "People would say that I have iron self-discipline") rated on a scale of 1 (not at all) to 7 (very much). A scale score was computed as the average of 13 items (after recoding reversed items), with higher scores representing greater self-control.

Participants then completed the Big Five Inventory[102], which consists of the stem "I am someone who…" followed by 44 items to assess conscientiousness (e.g., 'does a thorough job'); extraversion (e.g., 'is talkative'); openness to experience (e.g., 'is ingenious, a deep thinker'); agreeableness (e.g., 'has a forgiving nature'), and neuroticism (e.g., 'worries a lot'). Each item was rated on a scale of 1 (Disagree strongly) to 5 (Agree strongly). Separate scores were calculated for each subscale by taking the average of the corresponding items.

To assess behavioral approach and inhibition, participants completed the BIS/BAS scale[103]. This scale consists of 24 items. Seven items assess the strength of the behavior inhibition system (BIS; e.g., 'Criticism or scolding hurts me quite a bit.'), and 13 items assess different aspects of the behavioral activation System (BAS), including BAS drive (four items, e.g., 'When I want something I usually go all-out to get it.'), BAS fun seeking (four items, e.g., 'I crave excitement and new sensations.'), and BAS reward responsiveness (5 items, e.g., 'It would excite me to win a contest.'). Reliabilities were operationalized as Cronbach's alpha, and were acceptable for each variable, ranging from 0.65 to 0.87 (see Table 1 for descriptive statistics).

*Goal setting*. As part of the survey, participants were asked to describe four goals that they planned to pursue over the coming year. In line with past research (e.g., refs. [80,104]), goals were described as follows: "Personal goals are projects and concerns that people think about, plan for, carry out, and sometimes (though not always) complete or succeed at. They may be more or less difficult to implement; require only a few or a complex sequence of steps; represent different areas of a person's life; and be more or less time consuming, attractive, or urgent." Participants were then asked to write down their most important personal goal that they planned to pursue over the coming year, followed by 3 other goals they planned to pursue. Examples of goals include "achieve a 3.5 GPA", "be more social", and "work out at the gym".

*Tasks*

Flanker task: Participants performed an arrow version of the flanker task in which five arrowheads were presented in white on a black screen and the stimulus arrays could be either compatible (<<<<<, >>>>>) or incompatible (<<><<, >><>>). Participants had to respond to the direction of the central arrowhead while ignoring the flanking arrows. The right and left-facing arrow keys were responded to the right and left arrow keys of a millisecond accurate QWERTY keyboard (Empirisoft DirectIN Millisecond Accurate Keyboard). Trials commenced with the presentation of a fixation cross (250 ms) that was followed by the presentation of a flanker target stimulus until response (max: 1000 ms). The target trial was followed by a blank screen for between 600 and 1000 ms before the start of the next trial. Participants performed a total of 420 trials with equal proportions of compatible and incompatible trials. Participants were given self-paced breaks after every 60 trials. Participants were instructed to respond as fast as possible. This task was programmed in MediaLab (v2012, Empirisoft, New York, NY).

Time estimation task: The time-estimation task was used to elicit the RewP. The task was programmed in E-Prime 2.0 (Psychology Software Tools, Pittsburgh, PA) and started with a fixation cross for 250 ms that was followed by a blank screen. The participants' task was to press the X key once they estimated that one second had elapses since the presentation of the initial fixation cross. External feedback was presented at a fixed interval of 2000 ms after the initial fixation cross. Correct feedback was provided visually by a plus sign in the center of the screen, and incorrect feedback was provided by a minus sign. Correct feedback was provided if

**Table 3 Time one individual difference measures predicting the experience sampling measures.**

| | Proportion of desires | | | | Desire strength | | | | Resistance strength | | | | Desire enactment | | | | Desire conflict with four personal goals | | | |
|---|---|---|---|---|---|---|---|---|---|---|---|---|---|---|---|---|---|---|---|---|
| | r | Cor p | LCI; UCI | BF$_{10}$ | r | Cor p | LCI; UCI | BF$_{10}$ | r | Cor p | LCI; UCI | BF$_{10}$ | r | Cor p | LCI; UCI | BF | r | Cor p | LCI; UCI | BF |
| Extraversion | −0.03 | 0.677 | −0.17; 0.11 | 0.10 | 0.14 | 0.055 | 0.00; 0.28 | 0.56 | −0.04 | 0.606 | −0.18; 0.11 | 0.10 | 0.07 | 0.328 | −0.07; 0.21 | 0.926 | −0.04 | 0.617 | −0.07; 0.21 | 0.11 |
| Agreeableness | 0.07 | 0.340 | −0.07; 0.21 | 0.14 | 0.23 | 0.001 | 0.09; 0.36 | 15.12 | −0.13 | 0.064 | −0.27; 0.01 | 0.49 | −0.03 | 0.712 | −0.17; 0.12 | 0.926 | −0.01 | 0.905 | −0.16; 0.14 | 0.09 |
| Conscientiousness | 0.07 | 0.333 | −0.07; 0.21 | 0.14 | 0.13 | 0.077 | 0.01; 0.27 | 0.43 | −0.14 | 0.050 | −0.28; 0.00 | 0.61 | −0.03 | 0.729 | −0.17; 0.12 | 0.926 | −0.15 | 0.044 | −0.29; 0.00 | 0.70 |
| Neuroticism | 0.08 | 0.287 | −0.07; 0.22 | 0.16 | −0.02 | 0.788 | −0.16; 0.12 | 0.09 | 0.11 | 0.129 | −0.03; 0.25 | 0.28 | −0.04 | 0.632 | −0.18; 0.11 | 0.926 | 0.22 | 0.003 | 0.07; 0.35 | 6.54 |
| Openness | 0.17 | 0.022 | 0.02; 0.30 | 1.21 | 0.13 | 0.073 | −0.01; 0.27 | 0.45 | −0.07 | 0.349 | −0.21; 0.07 | 0.14 | 0.02 | 0.810 | −0.13; 0.16 | 0.926 | −0.24 | 0.001 | −0.37; −0.10 | 15.86 |
| TSC | 0.03 | 0.667 | −0.11; 0.17 | 0.10 | 0.10 | 0.175 | −0.04; 0.24 | 0.20 | −0.16 | 0.028 | −0.29; −0.02 | 1.00 | 0.02 | 0.801 | −0.12; 0.16 | 0.926 | 0.15 | 0.034 | −0.30; −0.01 | 0.87 |
| BIS | 0.09 | 0.201 | −0.05; 0.23 | 0.17 | 0.10 | 0.205 | −0.04; 0.24 | 0.17 | 0.06 | 0.413 | −0.08; 0.20 | 0.13 | −0.06 | 0.442 | −0.20; 0.09 | 0.926 | 0.15 | 0.141 | 0.01; 0.29 | 0.72 |
| BAS | 0.08 | 0.261 | −0.06; 0.22 | 0.17 | 0.24 | <0.001 | 0.11; 0.37 | 29.24 | −0.08 | 0.294 | −0.22; 0.07 | 0.16 | 0.02 | 0.765 | −0.12; 0.16 | 0.926 | −0.03 | 0.688 | −0.18; 0.12 | 0.10 |
| Early LPP high arousal pleasant | −0.08 | 0.292 | −0.22; 0.07 | 0.16 | 0.01 | 0.938 | −0.14; 0.15 | 0.09 | −0.05 | 0.520 | −0.19; 0.10 | 0.11 | −0.11 | 0.145 | −0.19; 0.10 | 0.580 | −0.10 | 0.210 | −0.24; 0.06 | 0.21 |
| LPP difference: high − low arousal | 0.04 | 0.642 | −0.11; 0.18 | 0.10 | −0.04 | 0.590 | −0.19; 0.11 | 0.11 | 0.05 | 0.526 | −0.10; 0.19 | 0.11 | −0.12 | 0.104 | −0.10; 0.19 | 0.580 | −0.03 | 0.702 | −0.18; 0.12 | 0.10 |
| LPP difference: positive − negative | −0.14 | 0.059 | −0.28; 0.01 | 0.55 | −0.11 | 0.159 | −0.25; 0.04 | 0.25 | −0.05 | 0.553 | −0.19; 0.10 | 0.09 | 0.01 | 0.952 | −0.14; 0.15 | 0.952 | −0.06 | 0.477 | −0.21; 0.10 | 0.12 |
| ΔERN | 0.06 | 0.488 | −0.10; 0.21 | 0.13 | 0.10 | 0.206 | −0.06; 0.25 | 0.22 | 0.03 | 0.697 | −0.13; 0.19 | 0.11 | −0.01 | 0.892 | −0.17; 0.15 | 0.951 | 0.07 | 0.372 | −0.09; 0.23 | 0.15 |
| ΔRewP | −0.07 | 0.380 | −0.21; 0.08 | 0.14 | 0.06 | 0.432 | −0.09; 0.20 | 0.13 | −0.07 | 0.368 | −0.21; 0.08 | 0.14 | 0.03 | 0.140 | −0.04; 0.25 | 0.580 | −0.13 | 0.107 | −0.27; 0.03 | 0.35 |
| RewP | −0.10 | 0.201 | −0.24; 0.05 | 0.21 | 0.10 | 0.181 | −0.05; 0.24 | 0.23 | −0.04 | 0.595 | −0.19; 0.11 | 0.11 | 0.06 | 0.418 | −0.09; 0.21 | 0.926 | −0.08 | 0.298 | −0.23; 0.07 | 0.17 |
| Flanker Δ RT | −0.07 | 0.389 | −0.21; 0.08 | 0.14 | 0.09 | 0.219 | −0.06; 0.24 | 0.20 | 0.04 | 0.650 | −0.12; 0.18 | 0.11 | 0.19 | 0.012 | 0.04; 0.33 | 0.192 | −0.03 | 0.741 | −0.18; 0.13 | 0.10 |
| error rate | −0.08 | 0.297 | −0.23; 0.07 | 0.16 | 0.04 | 0.639 | −0.11; 0.18 | 0.16 | 0.11 | 0.170 | −0.05; 0.25 | 0.24 | −0.02 | 0.773 | −0.17; 0.13 | 0.926 | −0.10 | 0.190 | −0.25; 0.05 | 0.23 |

Statistics presented in bold were significant after correction for multiple comparisons.
r Pearson's correlation coefficient (two-sided), p uncorrected p-value, Cor p corrected p-value using false discovery rate method, LCI lower 95% confidence interval, UCI upper 95% confidence interval.

the participants' response was within a pre-defined window that was centered around 1 s after the initial fixation cross. The duration of this window was titrated adaptively throughout performance to ensure that participants received roughly equal numbers of correct and incorrect feedback. The window was initially set at 100 ms, and was reduced or increased by 10 ms for accurate and inaccurate responses, respectively. This adaptive procedure meant that participants received roughly equal numbers of incorrect feedbacks. Participants first completed 20 practice trials of this task, followed by 168 experimental trials. The experimental trials were further divided into four blocks of equal length, separated by self-paced breaks.

Picture viewing task: To obtain the LPP, participants viewed 210 images (presented in random order): 30 each of negative and positive high arousal and low arousal that were analyzed here (full image list: https://osf.io/c283f/), as well as 30 neutral images from the IAPS that were not used in the current analyses (using the same materials and protocol as in ref. [105]). An additional 60 images were included to assess neural responses to healthy and unhealthy foods. Images were taken from the food-pics database (a large database of food and control images rated on characteristics such as valence and palatability, as well as micronutrient information; ref. [106]); as these food images were not relevant to the current paper, we will not consider them further. This task was programmed in MediaLab (v2012, Empirisoft, New York, NY).

*EEG measures.* EEG activity was recorded continuously throughout the entire in-lab session as the participants completed each task. The EEG was recorded from 36 Ag/AgCl sintered electrodes arranged according to the international 10–20 system in a stretch-lycra cap (Electro-Cap International, Eton, OH). Vertical electro-oculography (VEOG) was recorded via a supra- to sub-orbital bipolar montage surrounding the right eye. Impedances were monitored during recording and kept at less than 5 kΩ. The continuous EEG signal was digitized at 512 Hz using ASA acquisition hardware and software (asalab 4.9.4 software, TMSi Refa8 device; Advanced Neuro Technology, Enschede, the Netherlands). Recordings used the average earlobe and forehead electrodes as reference and ground, respectively. All subsequent data analyses were conducted offline using Brain Vision Analyzer (v.2.2; Brain Products, GmbH, Gilching, Germany). The offline EEG processing was pre-registered before our analyses began (https://osf.io/znsw8/).

The data were band-pass filtered offline using zero phase shift Butterworth filters (24 dB/octave roll-off) with a high-pass filter of 0.1 Hz and a low-pass filter of 20 Hz. Eye-blinks were corrected for using regression-based procedures[107]. Semiautomatic procedures were then used to identify and reject EEG artifacts. The artifact criteria were a voltage step of more than 25 μV between sample points, a voltage difference of 150 μV within 200 ms intervals, voltages above 85 μV and below −85 μV, and a maximum voltage difference of less than 0.05 μV within 100 ms intervals. Intervals were rejected on an individual channel basis to maximize data retention for the subsequent ERP analyses. Averaged ERPs were rejected if they comprised fewer than 5 trials for the response-locked ERPs, and fewer than 20 trials for the RewP, and fewer than 8 trials for the LPP (see https://osf.io/g759u/).

Reliability was assessed for each ERP score using split-half reliability assessment. This was achieved by ordering all viable epochs at the electrode of interest and creating separate averages for odd and even trials. A reliability statistic was then calculated by first conducting a Pearson's correlation between the odd and even ERPs (e.g., $r_{odd-even}$) and applying Spearman-Brown correction, $r_{SB} = 2*(r_{odd-even})/(1+r_{odd-even})$, to adjust for the smaller number of trials per condition as a result of creating the split-halves of the data [see ref. [108]]. Reliability for the difference waves was calculated by first subtracting the odd/even ERN split-half averages from the odd/even CRN split-half averages, and then subjecting these to the same analysis steps as the ERN/CRN to compute reliability.

The ERPs were then operationalized as follows. For the response-locked ERPs, epochs were created that started 400 ms before each response and lasted for 1400 ms. Epochs were then averaged separately for error and correct trials for each participant. The peak of the ERN was then operationalized by first creating a grand average ERN across all participants to find the peak ERN amplitude at electrode FCz. A 50 ms window surrounding this peak (26–76 ms) was then used to obtain a mean amplitude measure of the ERN (reliability = 0.86) and its correct-trial equivalent, the correct-related negativity (CRN, reliability = 0.98) for each participant. The difference ERN (ΔERN, reliability = 0.89, see Table 1) was obtained by subtracting error epochs from correct epochs and then extracting a mean amplitude measure in the same time-window used for the ERN. The ΔERN was used for our primary confirmatory tests and are presented in the main manuscript; however, we also present results for the CRN and ERN in the Supplementary Information (see Table S3). Importantly, the choice of ERP did not influence the conclusions drawn in the manuscript.

Feedback-related ERPs were identified using epochs that started 400 ms before a feedback stimulus and lasted for 1400 ms. These epochs were baseline corrected using a 200 ms window that started 200 ms before the onset of the feedback stimulus. The RewP was operationalized as a 50 ms mean amplitude window at FCz that was identified using a collapsed localizer method that collapsed across condition (i.e., correct feedback, error feedback) and participant. We extracted mean amplitude measures 255–305 ms after feedback onset separately for correct

**Table 4 Time one measures predicting longitudinal goal progress 1, 3, and 6 months later.**

| | Goal progress at 1 month (n = 149) | | | | | | Goal progress at 3 months (n = 132) | | | | | | Goal progress at 6 months (n = 107) | | | | | |
|---|---|---|---|---|---|---|---|---|---|---|---|---|---|---|---|---|---|---|
| | r | p | Cor p | LCI | UCI | BF | r | p | Cor p | LCI | UCI | BF | r | p | Cor p | LCI | UCI | BF |
| Extraversion | **0.21** | **0.011** | **0.044** | **0.05** | **0.36** | **2.43** | 0.16 | 0.075 | 0.300 | −0.02 | 0.32 | 0.52 | 0.03 | 0.741 | 0.935 | −0.16 | 0.22 | 0.13 |
| Agreeableness | **0.29** | **<0.001** | **<0.001** | **0.13** | **0.43** | **56.41** | 0.15 | 0.098 | 0.314 | −0.03 | 0.31 | 0.42 | **0.26** | **0.006** | **0.048** | **0.08** | **0.43** | **4.67** |
| Conscientiousness | 0.16 | 0.054 | 0.144 | 0.00 | 0.31 | 0.64 | 0.08 | 0.389 | 0.692 | −0.10 | 0.24 | 0.16 | 0.18 | 0.068 | 0.218 | −0.01 | 0.36 | 0.63 |
| Neuroticism | **−0.38** | **<0.001** | **<0.001** | **−0.51** | **−0.24** | **10697.68** | **−0.26** | **0.003** | **0.048** | **−0.41** | **−0.09** | **9.84** | −0.12 | 0.209 | 0.557 | −0.31 | 0.07 | 0.26 |
| Openness | 0.11 | 0.175 | 0.311 | −0.05 | 0.27 | 0.26 | 0.01 | 0.94 | 0.995 | −0.16 | 0.18 | 0.11 | 0.00 | 1.000 | 1.000 | −0.19 | 0.19 | 0.12 |
| TSC | **0.23** | **0.006** | **0.032** | **0.07** | **0.37** | **4.57** | 0.22 | 0.012 | 0.096 | 0.05 | 0.38 | 2.43 | **0.28** | **0.004** | **0.048** | **0.09** | **0.44** | **7.59** |
| BIS | −0.17 | 0.038 | 0.122 | −0.32 | −0.01 | 0.85 | −0.16 | 0.069 | 0.300 | −0.32 | 0.01 | 0.56 | 0.03 | 0.760 | 0.935 | −0.16 | 0.22 | 0.13 |
| BAS | 0.08 | 0.337 | 0.490 | −0.08 | 0.24 | 0.16 | −0.03 | 0.755 | 0.942 | −0.20 | 0.14 | 0.11 | 0.00 | 0.969 | 1.000 | −0.19 | 0.19 | 0.12 |
| Early LPP high arousal pleasant | 0.13 | 0.113 | 0.226 | −0.03 | 0.29 | 0.36 | 0.05 | 0.601 | 0.874 | −0.13 | 0.22 | 0.13 | 0.07 | 0.483 | 0.773 | −0.13 | 0.26 | 0.16 |
| LPP difference: high − low arousal | −0.09 | 0.284 | 0.454 | −0.25 | 0.08 | 0.19 | −0.14 | 0.128 | 0.341 | −0.31 | 0.04 | 0.35 | −0.24 | 0.017 | 0.091 | −0.41 | −0.04 | 2.02 |
| LPP difference: positive − negative | 0.07 | 0.397 | 0.529 | −0.09 | 0.23 | 0.15 | 0.10 | 0.286 | 0.572 | −0.08 | 0.27 | 0.20 | 0.10 | 0.302 | 0.690 | −0.09 | 0.29 | 0.21 |
| ΔERN | 0.06 | 0.468 | 0.544 | −0.11 | 0.23 | 0.14 | 0.07 | 0.471 | 0.754 | −0.12 | 0.25 | 0.15 | 0.08 | 0.433 | 0.773 | −0.12 | 0.28 | 0.18 |
| ΔRewP | 0.05 | 0.562 | 0.599 | −0.12 | 0.21 | 0.12 | −0.03 | 0.765 | 0.942 | −0.20 | 0.15 | 0.12 | −0.02 | 0.864 | 0.987 | −0.21 | 0.18 | 0.13 |
| RewP | 0.06 | 0.476 | 0.544 | −0.11 | 0.22 | 0.14 | 0.00 | 0.993 | 0.995 | −0.18 | 0.18 | 0.11 | 0.04 | 0.685 | 0.935 | −0.16 | 0.23 | 0.14 |
| Flanker Δ RT | −0.04 | 0.627 | 0.627 | −0.21 | 0.13 | 0.12 | −0.10 | 0.275 | 0.572 | −0.27 | 0.08 | 0.21 | −0.08 | 0.443 | 0.773 | −0.27 | 0.12 | 0.17 |
| Flanker Δ error rate | −0.15 | 0.079 | 0.181 | −0.31 | 0.02 | 0.49 | 0.00 | 0.995 | 0.995 | −0.18 | 0.18 | 0.11 | −0.19 | 0.054 | 0.216 | −0.38 | 0.00 | 0.78 |

Statistics presented in bold were significant after correction for multiple comparisons.
r Pearson's correlation coefficient (two-sided), p uncorrected p-value, Cor p corrected p-value using false discovery rate method, LCI lower 95% confidence interval, UCI upper 95% confidence interval.

(reliability = 0.97) and error (reliability = 0.95) trials. We used a difference wave (correct *minus* error; ΔRewP, reliability = 0.79) as our primary measure for confirmatory analyses and we also report the RewP to correct feedback in all tables (see Supplementary Information Table S3 for results with the RewP to error feedback). As with the ERN results, the choice of ERP operationalization did not influence the conclusions of our study.

The LPP was also operationalized in a window that started 400 ms before each emotional image and lasted for 2400 ms. The LPP was extracted as two adjacent 500 ms mean amplitude windows to reflect the early and late aspects of this components (early LPP: 350–850 ms; late LPP: 850–1350 ms). As with the Pe and RewP, these aspects of the LPP were operationalized using a collapsed localized method across conditions and participants at electrode Pz. Our analyses focused on three operationalizations of the LPP. First, we analyzed initial orienting to the high arousal positive stimuli using the early LPP to high arousal positive images (split-half reliability = 0.82; see Table 1). We additionally computed difference waves (e.g., positive *minus* negative; high arousal minus low arousal), see Figs. 3 and 4. However, these variables demonstrated poor split-half reliability (ranging from 0.23 to 0.54) and should be treated with caution in the subsequent analyses. We also present supplementary ERP analyses for the early and late LPP to every image type in the Supplementary Information (see Table S3). These analyses supported the same conclusions as those in the main manuscript.

*Experience sampling.* In the experience sampling survey, participants were first asked about whether they were currently experiencing a desire or had experienced one in the past 30 min. When participants indicated that they were or had recently experienced a desire, they then indicated what the desire was for, choosing from among 23 categories (adapted from ref. [82]; see all materials on OSF). They then reported on desire strength ('how strong is/was the desire?') using a slider scale ranging from 1 (very weak) to 7 (very strong) and whether they had the opportunity to satisfy the desire (y/n). If they indicated that they had the opportunity to satisfy the desire, they were asked about resistance ('did you try to resist the desire'), using a slider with the anchors 1 (did not try to resist at all) and 7 (tried very hard to resist). Those who reported resisting at least somewhat (did not select 1) were asked about the strategies that they used to resist. Participants then reported whether they gave in to the desire (y/n). Please note that we had initially proposed to operationalize desire strength using a variable that accounts for instances where participants did not experience a desire. We created a variable where all reports of desire strength for experienced desires were copied over, and instances where participants did not experience a desire were recorded as 0 on this new variable (so that the new desire strength variable now ranged from 0 to 7). However, this variable was almost redundant with proportion of desires (r = 0.92), so we kept the original measure of desires (departing from our pre-registration).

For supplemental (non-pre-registered) analyses, we also examined the extent to which the desires conflicted with or helped four personal goals that participants reported at baseline. Ratings for each goal ranged from −3 (conflicts with goal pursuit) to 3 (helps with goal pursuit); a measure of conflict strength was computed for each desire by averaging across the four goals and reverse-scoring it (such that larger numbers indicated greater conflict). We also computed additional measures of desire strength, resistance strength, and desire enactment for only those desires that conflicted at least somewhat with at least one of the goals; results with these variables are reported in the Supplementary Information (see Table S2).

Follow-up goal progress: At each of three follow-ups (1, 3, and 6 months later), participants were reminded of their four goals that they set at the baseline assessment and asked about goal progress using 3 items previously used in goal research (e.g., "I have made a lot of progress towards this goal"[10,104]) rated on a scale of 1 (strongly disagree) to 7 (strongly agree). The average across the three items, and the four goals, at each time constituted the goal progress variable.

Supporting the psychometric validity of these measures, the internal consistency for each of these items (12 items per time point, 3 items for each of the 4 goals) was high (Cronbach's α = 0.83 at 1 month, 0.80 at 3 months, and 0.84 at 6 months). As noted in Table 2, we also found that goal progress measures were correlated positively with each other (rs = 0.37–0.53), suggesting that self-reported goal progress was moderately stable over the 6-month period.

**Planned analyses.** Given that here we were interested in person-level variables, we aggregated our key desire variables by person, computing for each person new variables representing the proportion of desires reported (out of all the signals that the person completed), the average desire strength, the average resistance strength, and the proportion of desires that were enacted (out of all the desires reported). We then correlated these variables with our personality and neural indices. Given that we conducted 16 correlations for each DV, a False Discovery Rate correction[109] was applied to reduce the proportion of false positives. In addition, to corroborate results, we used Bayesian analyses to compute a Bayes factors for each correlation (using JASP software; 0.14.1). The Bayesian Pearson's correlation analyses used default settings in JASP to test for an alternative hypothesis that variable pairs were correlated using a stretched beta prior width of 1.0.

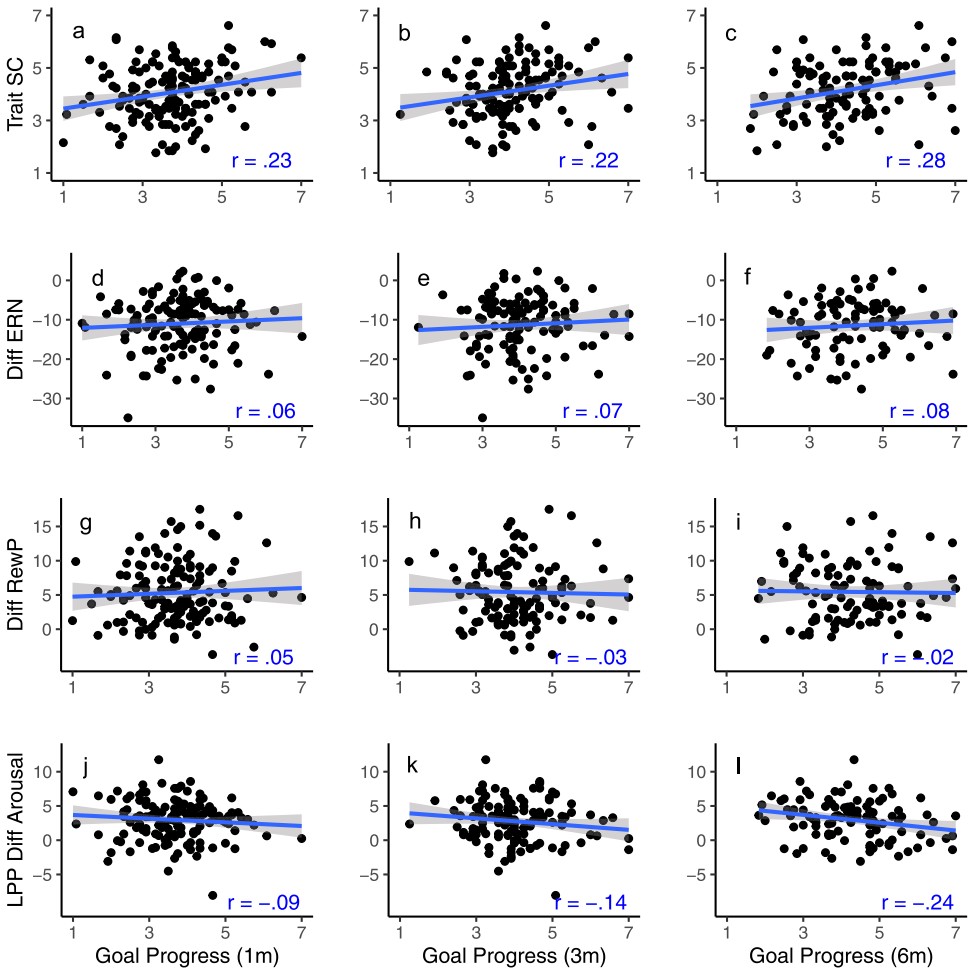

**Fig. 6 Scatterplots showing Pearson's correlation coefficients between the key individual difference predictors and longitudinal goal progress at 1-, 3-, and 6-month follow-ups.** The relationship between goal progress at 1, 3, and 6 months are depicted in separate plots for trait self-control (**a–c**), the difference ERN amplitude (**d–f**), the difference RewP (**g–i**), and the LPP difference between arousal levels (**j–l**). Lines depict simple linear regression to aid visual interpretation, error bands depict the standard error. Trait SC: trait self-control; ERN: error-related negativity; RewP: reward positivity; LPP: late positive potential.

**Reporting summary**. Further information on research design is available in the Nature Research Reporting Summary linked to this article.

## Data availability
The data underlying our findings are available under restricted access due to ethical restrictions, access can be obtained by emailing the corresponding authors.

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

## Acknowledgements

This research was supported by grants from the Social Sciences and Humanities Research Council of Canada to Marina Milyavskaya (#435-2016-0991) and to Michael Inzlicht (#435-2019-0144), as well as a grant from the Natural Sciences and Engineering Research Council of Canada to Michael Inzlicht (RGPIN-2019-05280).

## Author contributions

B.S., M.M., and M.I. designed the study. M.M. and B.S. oversaw data collection, analyzed the data, and co-wrote the manuscript. B.S. created the visualizations. M.I. provided feedback on analyses and edited the manuscript.

## Competing interests

The authors declare no competing interests.
