## [Peer Review File · Nature Communications]

Longitudinal evidence that Event Related Potential measures of self-regulation do not predict everyday goal pursuitEditorial Note: This manuscript has been previously reviewed at another journal that is not operating a transparent peer review scheme. This document only contains reviewer comments and rebuttal letters for versions considered at Nature Communications.

REVIEWER COMMENTS

Reviewer #1 (Remarks to the Author):

The authors have successfully addressed all my comments. I particularly appreciate the thoughtful (and more balanced) considerations of the study's implications for cognitive neuroscience theories. I have a few remaining comments which would require only minor to moderate revisions.

Another possibility for the observed null findings is that the ERN might be a good state measure but a bad trait measure of control/self-regulation. In a recent study, Fischer et al. (2016) have shown in a sample with 874 subjects that the ERN predicts post-error slowing on a trial-by-trial basis but that both measures do not correlate across subjects. This could suggest that the source of within-subjects variability (that is typically used for testing cognitive neuroscience models of control) and the source of between-subjects variability (that is used to predict self-regulation) differ in some respect, possibly because between-subjects variability reflects to a greater extent task-specific variables (how easily can I detect errors in this specific task?) or neuroanatomical variables. This is mere speculation but it might point to the core of the problem.

Line 277: The term „feedback accuracy“ is somewhat misleading here as the variable does not refer to the accuracy of the feedback but to the accuracy of the response. „Feedback valence“ might be a better choice.

Line 432-433: „... raise the possibility that the ERN might not be particularly relevant for everyday goal pursuit, ...“

This formulation sounds too strong. First, the fact that "the ERN in a a lab-based task is not predictive of everyday goal pursuit" can not be generalised to "any kind of ERN (including those that can potentially be measured during everyday behaviour) is not predictive of everyday goal pursuit". Second, even if we were able to measure an ERN during everyday goal pursuit and even if we would replicate the present findings in this context, we would probably not conclude that the ERN is irrelevant here - it would at least indicate that errors are detected.

Ref

Fischer, A. G., Danielmeier, C., Villringer, A., Klein, T. A., & Ullsperger, M. (2016). Gender influences on brain responses to errors and post-error adjustments. *Scientific reports*, 6, 24435.

Reviewer #2 (Remarks to the Author):

The authors provided a thoughtful and thorough revision, however I do not think that it successfully addressed my major concerns with the report. Namely, the following concerns about the validity of the findings are compromised by conceptual incongruity (linking individual differences in systems-level measures to a highly abstract and long time-frame outcome like goal attainment) as well as to a smaller degree methodological concerns:

1) Low power (only 80% sensitive to r 's > .23 (line 494))

2) A vague and highly derived construct used as the dependent variable, which limits the power and precision of statistical tests ("e.g. I have made a lot of progress towards this goal"; ... rated on a scale of 1 (strongly disagree) to 7 (strongly agree). The average across the three items, and the four goals, at each time constituted the goal progress variable")

Response to reviewer comments for “Self-regulation in the lab does not predict goal progress in the wild: Longitudinal evidence from ERPs, experience sampling, and goal progress” (NCOMMS-21-37775-T).

Reviewer #1 (Remarks to the Author):

“The authors have successfully addressed all my comments. I particularly appreciate the thoughtful (and more balanced) considerations of the study’s implications for cognitive neuroscience theories. I have a few remaining comments which would require only minor to moderate revisions.”

Thank you for this comment, we really appreciated the previous round of reviews and their impact on sharpening the contribution of our paper.

“Another possibility for the observed null findings is that the ERN might be a good state measure but a bad trait measure of control/self-regulation. In a recent study, Fischer et al. (2016) have shown in a sample with 874 subjects that the ERN predicts post-error slowing on a trial-by-trial basis but that both measures do not correlate across subjects. This could suggest that the source of within-subjects variability (that is typically used for testing cognitive neuroscience models of control) and the source of between-subjects variability (that is used to predict self-regulation) differ in some respect, possibly because between-subjects variability reflects to a greater extent task-specific variables (how easily can I detect errors in this specific task?) or neuroanatomical variables. This is mere speculation but it might point to the core of the problem.”

Thank you for bringing this within vs. between subject distinction to our attention. We agree that this is a potentially generative line of reasoning for future studies that might look at within-subject analyses (this was not possible with our current design). We have adapted the discussion section of our manuscript as follows:

“It is worth noting that individual differences in ERN amplitudes often fail to predict between subject variation during the performance of laboratory cognitive control tasks themselves, such as slowing after initial mistakes to become more cautious (i.e., posterror slowing, Fischer et al., 2016). These findings, in addition to our own, suggest that the ERN might serve poorly as an individual difference predictor of subsequent behaviour—even when this behavioural adjustment and the ERN arise within the context of the same task. In contrast, within-person variation in ERN amplitude does predict within-person adjustments in response caution on a trial-by-trial basis (Cavanagh & Shackman, 2015; Fischer et al., 2016). Future studies might test this idea outside of the lab by, for example, pairing ambulatory assessments of the ERN with experience sampling to test if within-person error-

related brain activity tracks with enhanced self-regulation on a moment-to-moment basis” (pp. 27-28).

“Line 277: The term „feedback accuracy“ is somewhat misleading here as the variable does not refer to the accuracy of the feedback but to the accuracy of the response. „Feedback valence“ might be a better choice.”

That is correct, we have now swapped the term ‘accuracy’ for ‘valence’ (p. 13).

“Line 432-433: „... raise the possibility that the ERN might not be particularly relevant for everyday goal pursuit, ...“

This formulation sounds too strong. First, the fact that "the ERN in a lab-based task is not predictive of everyday goal pursuit" can not be generalised to "any kind of ERN (including those that can potentially be measured during everyday behaviour) is not predictive of everyday goal pursuit". Second, even if we were able to measure an ERN during everyday goal pursuit and even if we would replicate the present findings in this context, we would probably not conclude that the ERN is irrelevant here - it would at least indicate that errors are detected.

We have now removed this phrasing from the end of this paragraph. We do believe that our work raises questions about the predictive validity of the ERN in the domain of goal pursuit, but we have now framed the end of this paragraph more generatively – pointing to studies that could continue to test the predictive validity of the ERN. See below:

“While no single study can provide a definitive answer to the broadest question of the general predictive validity of the ERN, our results (in addition to considerations of the likely low statistical power of past investigations; Pavlov et al., 2021) raise questions about the ability of lab-derived ERN scores to predict the longer-term pursuit of personal goals. Future work would benefit from conducting larger scale confirmatory attempts to replicate previously identified ERN-outcome associations in other, perhaps more specific, domains (e.g., between the ERN and academic grades, or between the ERN and everyday emotion regulation)” (p. 27).

Reviewer #2 (Remarks to the Author):

“The authors provided a thoughtful and thorough revision, however I do not think that it successfully addressed my major concerns with the report. Namely, the following concerns about the validity of the findings are compromised by conceptual incongruity (linking individual differences in systems-level measures to a highly abstract and long time-frame outcome like goal attainment) as well as to a smaller degree methodological concerns:”

Thank you for reviewing our manuscript once more, and we appreciate these comments and critique. We have addressed the specific concerns below.

“1) Low power (only 80% sensitive to r 's > .23 (line 494))”

We do not agree that these characteristics are necessarily indicative of ‘low power’ in an absolute sense. An r of $\sim .2$ is commonly considered as a small effect size, and our sensitivity analyses indicate that we have 80% power to detect an effect size at this level. This gives us a good chance to find effect sizes that are considered typically small in psychology. Indeed, our study—we believe—is uniquely large among longitudinal EEG studies and is larger than most individual difference studies using ERPs (cf., Pavlov et al., 2021, *Cortex*). We also combined these longer-term follow-ups with a week-long intensive longitudinal experience sampling study, further increasing the rigour of our design to find correlations between the lab and real-world self-regulation.

Nevertheless, it is true that we could always increase statistical power, and we think this will be an important (and significant) challenge for ongoing individual difference research across neuroscience methods. We have included a discussion of power and its limitations in the discussion section:

“Additionally, even though experience sampling results in many observations per participant, here we aggregated these observations to examine individual differences (i.e., in how they generally feel and act across the week). This comes at a cost to power. Although our sample sizes are high for an EEG study, we only had 80% power to detect correlations greater than $r = .23$ at the one-month follow-up, rising to $r = .27$ at six months. As such, our study meets common power conventions to detect small effect sizes (Cohen, 1992), but has lower power to detect even smaller effects. It is important to note that our Bayesian analyses did provide evidence in favour of null relationships—suggesting that samples were not too small to provide evidence in relation to our hypotheses. However, these considerations highlight that potentially even larger, preregistered studies are required to continue to investigate the predictive validity of ERPs for everyday outcomes. Increased statistical power will not, however, overcome the possibility (as discussed above) that individual differences extracted from behavioural cognitive control tasks, including their neural correlates, are not particularly valid predictors of everyday self-regulation” (pp. 29-30).

“2) A vague and highly derived construct used as the dependent variable, which limits the power and precision of statistical tests (“e.g. I have made a lot of progress towards this goal”; ... rated on a scale of 1 (strongly disagree) to 7 (strongly agree). The average across the three items, and the four goals, at each time constituted the goal progress variable”)

We understand the reviewer's concerns and have now included further analyses of these outcome measures in the revised manuscript. Because we studied personal goals in our study, it was necessary to have relatively open self-report measures of goal progress so that these would be relevant regardless of the goal nominated by the participant at time one. Note that these items are frequently used in research on personal (idiosyncratic) goal progress, across many different labs (e.g., Benita et al., 2021; Koestner et al., 2008; Werner & Milyavskaya, 2018). Moreover, past research has found that these sorts of goal progress measures covary with other measures in theoretically meaningful ways (e.g., Milyavskaya & Inzlicht, 2017). We have now included psychometric details for these scales (as we had with other measures) to verify that they are internally consistent— we observed Cronbach's alpha of .8 or more for these measures at each time-point. We also note that goal progress at each timepoint was positively correlated, and that this goal progress measure has been used in previous studies and is related to theoretically related constructs (i.e., trait self-control; see discussion pp. 28-29). For this reason, we think that there is good support for the validity of our goal outcome measures.

Details on the psychometric properties of these measures appears in the paper as follows:

“Supporting the psychometric validity of these measures, the internal consistency for each of these items (12 items per time point, 3 items for each of the 4 goals) was high (Cronbach's $\alpha = .83$ at 1 month, $.80$ at 3 months, and $.84$ at 6 months). As noted in Table 2, we also found that goal progress measures were correlated positively with each other ($r_s = .37-.53$), suggesting that self-reported goal progress was moderately stable over the 6-month period.” (pp. 39-40)

Once again, I would like to thank the reviewers for your constructive evaluations of our manuscript. We are happy to make any further revisions that may be required,

Yours sincerely,
Blair Saunders

REVIEWERS' COMMENTS

Reviewer #1 (Remarks to the Author):

The authors have adequately responded to my comments. I have no further concerns.